# An Efficient Deep Learning Approach for Colon Cancer Detection

Ahmed S. Sakr [1], Naglaa F. Soliman [2,*], Mehdhar S. Al-Gaashani [3], Paweł Pławiak [4,5], Abdelhamied A. Ateya [6] and Mohamed Hammad [7,*]

1   Department of Information System, Faculty of Computers and Information, Menoufia University, Shibin El Kom 32511, Egypt
2   Department of Information Technology, College of Computer and Information Sciences, Princess Nourah bint Abdulrahman University, P.O. Box 84428, Riyadh 11671, Saudi Arabia
3   College of Computer Science and Technology, Chongqing University of Posts and Telecommunications, Chongqing 400065, China
4   Department of Computer Science, Faculty of Computer Science and Telecommunications, Cracow University of Technology, Warszawska 24, 31-155 Krakow, Poland
5   Institute of Theoretical and Applied Informatics, Polish Academy of Sciences, Bałtycka 5, 44-100 Gliwice, Poland
6   Department of Electronics and Communications Engineering, Zagazig University, Zagazig 7120001, Egypt
7   Department of Information Technology, Faculty of Computers and Information, Menoufia University, Shibin El Kom 32511, Egypt
*   Correspondence: nfsoliman@pnu.edu.sa (N.F.S.); mohammed.adel@ci.menofia.edu.eg (M.H.)

**Abstract:** Colon cancer is the second most common cause of cancer death in women and the third most common cause of cancer death in men. Therefore, early detection of this cancer can lead to lower infection and death rates. In this research, we propose a new lightweight deep learning approach based on a Convolutional Neural Network (CNN) for efficient colon cancer detection. In our method, the input histopathological images are normalized before feeding them into our CNN model, and then colon cancer detection is performed. The efficiency of the proposed system is analyzed with publicly available histopathological images database and compared with the state-of-the-art existing methods for colon cancer detection. The result analysis demonstrates that the proposed deep model for colon cancer detection provides a higher accuracy of 99.50%, which is considered the best accuracy compared with the majority of other deep learning approaches. Because of this high result, the proposed approach is computationally efficient.

**Keywords:** CNN; colon cancer; deep learning; histopathological images; lightweight model

## 1. Introduction

Adenocarcinomas, which damage the lining of the large intestine (colon) and rectum, account for nearly all malignancies of the large intestine and rectum large intestine and rectum malignancies. Colorectal cancer typically originates as a button-like growth termed a polyp on the surface of the intestinal lining or rectum [1]. The intestine or rectum division, may invade nearby or adjacent lymph nodes. Due to the fact that blood flows from the intestine's wall and a substantial portion of the rectum to the liver, colorectal cancer can metastasize to the liver after spreading to adjacent lymph nodes [2]. Cancer of the large intestine and rectum is the most frequent cancer in Western countries and the second most significant cause of cancer death. When a person is between 40 and 50 years, their risk of acquiring colorectal cancer significantly increases. Around 140,250 persons are affected by colon and rectal cancer. Each year, approximately 50,630 people die in the United States [3]. Colorectal cancer risk is increased by family history and specific dietary factors (low fiber and high fat) [4]. Typical symptoms include blood in feces, fatigue, and weakness. As a result, cancer that is caught early is more curable. In this paper, we propose a novel and efficient system for the automatic early detection of colon cancer from the analysis of histopathological images.

Recently, several computer-assisted diagnosis systems (CADs) were introduced to automatically check for signs of tumor or cancer growth in the colon [5–15]. Thanks to artificial intelligence (AI) for helping the diagnostic systems accurately detect cancer, the common AI approaches for cancer detection are: (1) detection of colon cancer using machine learning approaches from the analysis of histopathological images [13–15] and (2) detection of colon cancer using deep learning approaches from the analysis of histopathological images [5–12].

In machine learning approaches, researchers used stages called preprocessing, feature extraction, feature selection, and optimization and finally classification stage. The preprocessing stage is used to remove any noise that effect on the accuracy of the original images. In the feature extraction stage, all unique features are extracted from the preprocessed image. After that, the most important features are selected from the extracted features in the feature selection stage. The final decision is made using a separate classifier in the classification stage. The recent works in this field using machine learning obtained a good performance for colon cancer detection. However, the cancer detection systems based on machine learning need manual detection of the features and separate classifiers for the detection, making the system more complex and time-consuming when using big data [16]. In addition, most of these methods suffer from overfitting problems and obtain low accuracies when working on big data. Finally, these methods depend on the extracted features and the quality of the feature extraction methods which affect the final classification results.

On the other hand, deep learning approaches alleviate the majority of the drawbacks associated with prior machine learning approaches [17]. In these approaches, the feature extraction and classification stages are combined in one stage using the deep model. The researchers employed the systems-based deep learning approaches usually used pretrained models and transfer learning techniques or by establishing their own deep model. The most common deep learning model is the convolutional neural network (CNN), which attained high accuracy for colon cancer detection. Several deep learning applications for medical image analysis, such as detection in 3D ultrasound images, COVID detection, and several types of cancer detection have been implemented in many works of literature [18–20]. These cancer detection systems based on deep learning often require big data. However, these methods obtained higher accuracies than machine learning methods on big data because of the deep methods' ability to extract powerful high-level features. In this study, we proposed a novel deep learning approach using CNN to detect of colon cancer. The main novel contributions of this study are:

- We propose a new end-to-end lightweight deep learning approach based on CNN for efficient colon cancer detection. Unlike the previous methods, the proposed method is less complex and consists of a few layers. In addition, unlike most previous work, our method is ended to end without using any external stages of machine learning such as feature extraction and classification stages. Our method outperformed most of the previous deep learning approaches in this field.
- The efficiency of the proposed system is analyzed with histopathological images database and is compared with the existing state-of-the-art methods in this field. Unlike the other methods, the proposed method achieved the highest accuracy using a small database.
- We propose an algorithm that can achieve good results at classification tasks, which is an important component for the development of automated computer-aided systems for colon cancer detection. Part of our source code can be found at:

The remainder of the paper is structured as follows. Section 2 discusses the recent related work in this field. Section 3 describes the proposed method based on a lightweight deep learning model for colon cancer detection. Section 4 explains the results and discussion where proposed approach is compared with the existing state-of-the-art approaches. Section 5 concludes the proposed research and, future enhancement is given.

## 2. Related Work

Recent literature mainly focused on using deep learning and transfer learning to detect colon cancer from the analysis of histopathological images [5–12]. Tongaçar [5] showed how to categorize lung and colon cancer using an AI-supported model and optimization methods using histopathological images. The researcher analyzed a set of five histopathological image classes, two for colon cancer and three for lung cancer. To train the image classes from scratch, the DarkNet-19 model was employed and then a support vector machine (SVM) was used to classify the features. Overall, the approach achieved a 99.69% accuracy rate. Kumar et al. [6] conducted a comparative investigation of two feature extraction approaches for categorizing of lung and colon cancers. They extracted six handcrafted traits based on color, texture, shape, and structure. They used multiple classifiers with handcrafted features for colon cancer classification. The extracted deep features classify lung and colon cancers using traditional classifiers. They obtained the best accuracy of 98.60%, recall of 98.60%, precision of 98.63% and F1 score of 0.985 using DenseNet-121 for feature extraction with RF classifier. Yildirim and Cinar [7], presented a method based on CNN to detect of colon cancer images. They used a 45-layer model in a model called MA_ColonNET to classify colon cancer. They attained a 99.75% accuracy rate, demonstrating that the presented approach can detect colon cancer earlier and facilitate treatment. Masud et al. [8] developed a categorization scheme for histopathological differentiating two benign and three cancerous lung and colon tissues. The researchers discovered that their framework is capable of accurately identifying cancer tissues to a maximum of 96.33%. They claimed that by implementing their methodology, medical experts would be able to establish an automated and reliable system for detecting various forms of lung and colon cancers.

## 3. Deep Learning Approach

In this section, we describe the dataset used in this study. This is followed by presenting an overview of the proposed colon cancer detection method.

### 3.1. Histopathological Image Data

We evaluated our work on the LC25000 Lung and colon histopathological image dataset [21]. This dataset contains 10,000 histopathological images of colon tissue classified into two classes such as benign colon (5000 images) and colon adenocarcinomas (5000 images). The size of each image is 768 × 768 pixels in JPEG format. We divided the data into 6400 for train, 1600 for validation and 2000 for test. Figure 1 shows visual examples of the dataset.

### 3.2. The Proposed Approach

The general schematic view of the proposed methodology is shown in Figure 2. This paper mainly aims to detect the colon cancer from the analysis of histopathological images. Initially, the input histopathological images are rescaling to 180 × 180, which decreasing the processing time. These images are then normalized before feeding them into our CNN model and colon cancer detection is performed. While developing our model, we used the validation split technique by dividing the training data to 80% images for train and 20% for validation. In this paper, we tested seven deep architectures with different batches and different layers and then selected the best performance architecture to be our model.

The first model consists of 14 layers with a total of 2,031,778 parameters. In this model, we employed one dropout layer with rate of 0.5% and batch size of 8. The second model consists of 16 layers with total parameters of 2,097,826 parameters. In this model, we employed the same dropout and batch size of the first model. The third model consists of 10 layers with total parameters of 7,954,210 parameters. This model is also employed the same dropout and batch size of the previous models. The fourth model consist of 14 layers with total parameters of 2,031,778 parameters. This model is different from the first model in the structure of layers and dropout rate which is 0.8%. The fifth model consist of 12 layers

with total parameters of 4,063,138 parameters. In this model, we employed the dropout layer with 0.8% and batch size of 8. The model number 6 consists of 10 layers with total parameters of 7,954,210 parameters. This model is different from the third model in the structure of layers and dropout rate which is 0.8%. The last model consists of 12 layers with total parameters of 4,063,138. This model is different from the third model in the structure of layers and dropout rate which is 0.5%. In this study, we selected the last model as the proposed model, which is achieved the highest accuracy compared to the other models.

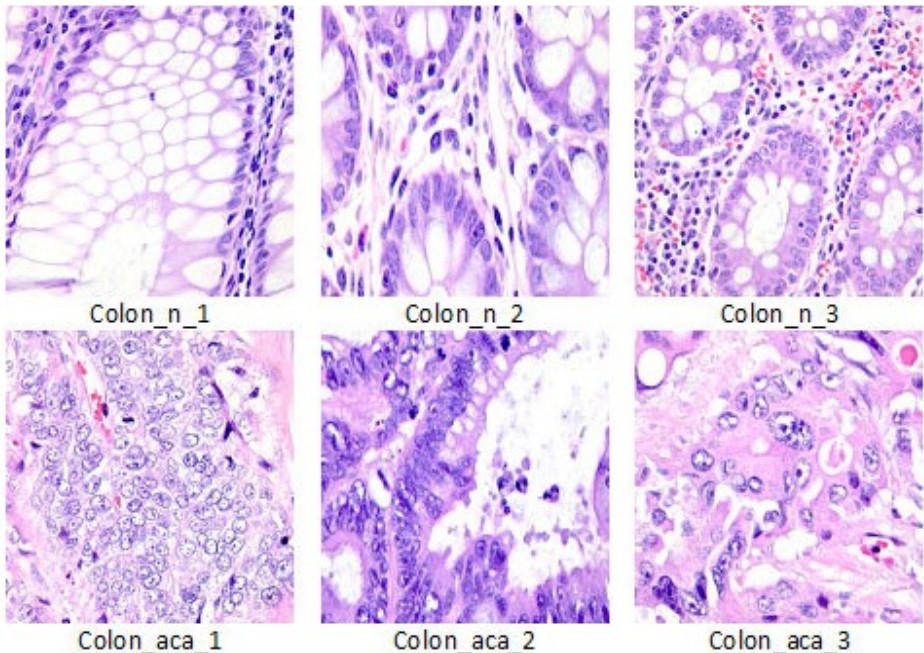

**Figure 1.** Visual examples of nine histopathological images from the used dataset (where colon_n refers to normal image and colon_aca refers to an image with colon cancer).

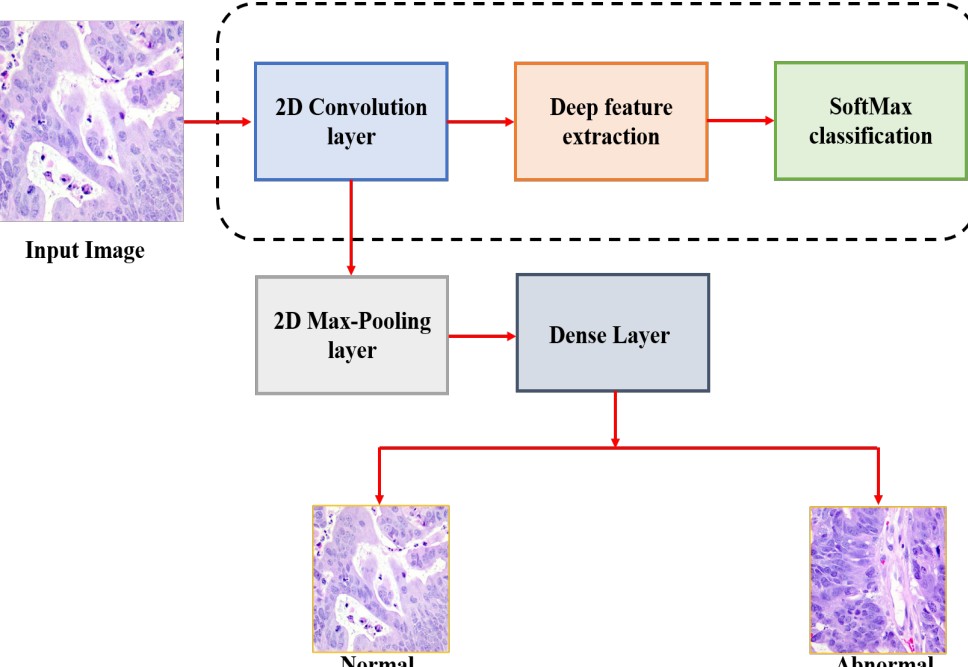

**Figure 2.** General Architecture of the Proposed Approach.

The Architecture of the CNN Model

The model is composed of four convolution blocks, each of which contains a max pool layer. There is a flatten layer on the bottom and a fully connected layer with 256 units on top that is activated by a ReLU activation function. In addition, the dropout layer is added after the flatten layer with a probability of 0.5 and finally SoftMax layer is added for classification. The Adam optimizer [22] and Cross entropy loss function [23] are used for model optimization. The model summary (the output shape of each layer and the parameter number) is shown in Figure 3. To train the model we used a minimum batch equal to 8 and the model stopped after 30 epochs with total of 4,063,138 parameters. The image batch represents the shape as a tensor (8, 180, 180, 3). This is a batch of eight images of the shape $180 \times 180 \times 3$. The overall architecture of our model with the data flow of each iteration in detail is shown in Figure 4.

```
Layer (type)                  Output Shape             Param #
=================================================================
 rescaling_1 (Rescaling)      (None, 180, 180, 3)      0

 conv2d (Conv2D)              (None, 180, 180, 16)     448

 max_pooling2d (MaxPooling2D  (None, 90, 90, 16)       0
 )

 conv2d_1 (Conv2D)            (None, 90, 90, 32)       4640

 max_pooling2d_1 (MaxPooling  (None, 45, 45, 32)       0
 2D)

 conv2d_2 (Conv2D)            (None, 45, 45, 64)       18496

 max_pooling2d_2 (MaxPooling  (None, 22, 22, 64)       0
 2D)

 conv2d_3 (Conv2D)            (None, 22, 22, 128)      73856

 max_pooling2d_3 (MaxPooling  (None, 11, 11, 128)      0
 2D)

 flatten (Flatten)            (None, 15488)            0

 dropout (Dropout)            (None, 15488)            0

 dense (Dense)                (None, 256)              3965184

 dense_1 (Dense)              (None, 2)                514

=================================================================
Total params: 4,063,138
Trainable params: 4,063,138
Non-trainable params: 0
```

**Figure 3.** Summary of our deep model (the output shape and number of parameters in each layer).

The first convolution layers are used to extract simple and obvious features (called feature map), such as edges in different directions. As we dig deeper into the hidden layers of the network, the complexity of the attributes that must be identified and extracted increases.

The purpose of the pooling layer is to reduce the size of the feature maps for reducing the amount of computation necessary and prevent the overfitting state. In our model, we employed max pooling to clear the feature maps in a small window and keep the larger values within each window and reducing the size of the map. Therefore, in the pooling layers, the best values for the attributes that resulted from the convolution layer are extracted.

The dense (SoftMax) layer is the final layer in our model, which contains neurons that are fully connected to all nodes from the preceding layer. In this layer, the final classification process occurs. A flattening operation is applied to simplify the input to a vector of attributes and then pass it to the dense layer to predict the output probabilities.

We employed the ReLU activation function in all layers to increase the non-linearity, mainly to solve the sigmoid function-induced gradient disappearance problem. In addition, dropout is used after the flattened layer to randomly exclude 50% of neurons in order to reduce the likelihood of overfitting.

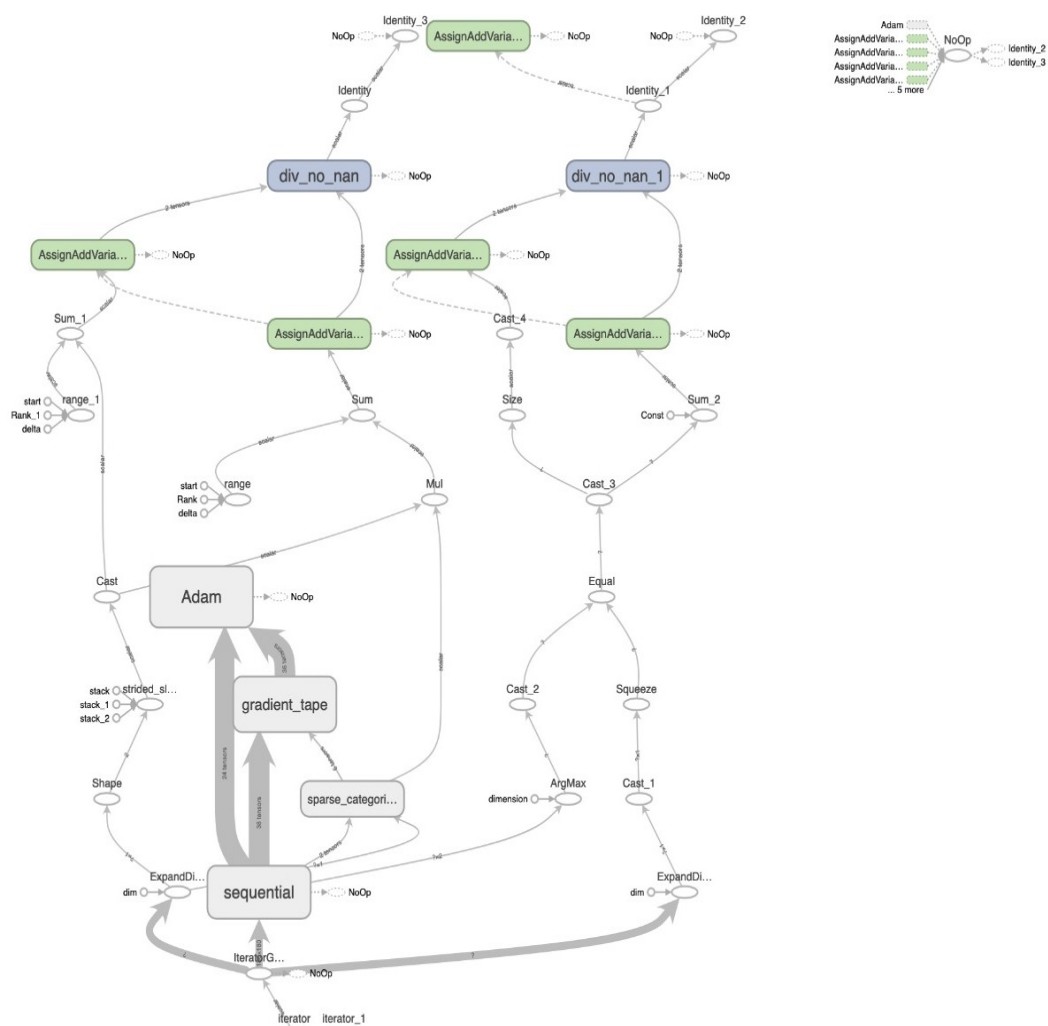

**Figure 4.** Data flow of our model.

## 4. Results and Discussion

The proposed method is tested on histopathological images collected from LC25000 dataset [21]. Python 3.7 is used in all simulations for histopathological images as well as for training datasets and testing datasets. The Cross-Entropy function was used during the training process to calculate the loss after a training iteration. In this work, we adopt accuracy, which is the most important performance measure, to analyze our scheme's generalization ability and classification performance. In addition, the confusion matrix is assumed to translate our binary classification according to True Positive (TP), False Positive (FP), True Negative (TN), and False Negative (FN). Moreover, recall, precision, and f1-score are computed to assess our results. The detailed definitions are given as follows [24]:

$$\text{precision} = \frac{TP}{(TP + FP)} \tag{1}$$

$$\text{recall} = \frac{TP}{(TP + FN)} \tag{2}$$

$$\text{Accuracy} = \frac{(TP + FN)}{(TP + TN + FP + FN)} \tag{3}$$

$$\text{f1-score} = \frac{TP}{(TP + \frac{1}{2}(FP + FN))} \tag{4}$$

$$SPE = \frac{TN}{TN + FP} \tag{5}$$

$$MCC = \frac{TP \times TN - FP \times FN}{\sqrt{(TP + FN) \times (TN + FN) \times (TP + FP) \times (TN + FP)}} \tag{6}$$

We tested the data using seven models and selected the best one to be our model. Table 1 shows a comparison between the seven models.

In Table 1, we can find that the best model is model 7, which achieved the highest accuracy compared with other models. Therefore, we selected model 7 as our model for all evaluations and the comparison with other previous models.

Figure 5 shows the confusion matrix of our method and the confusion matrixes of other compared models. From the figure, we can show that our model can detect all abnormal images correctly with an accuracy of 100% and the proposed model can accurately detect 99% of the normal images and detect 1% of the normal images as abnormal images from the used dataset. Figure 6 shows the accuracy and loss curves for the training and validation data during 30 epochs. From the figure, we can show that, after 10 epochs of training, the improvements are marginal and after 15 epochs, the model can be considered fully trained.

The overall performance of the proposed CNN model based on the presented evaluation metrics is shown in Table 2.

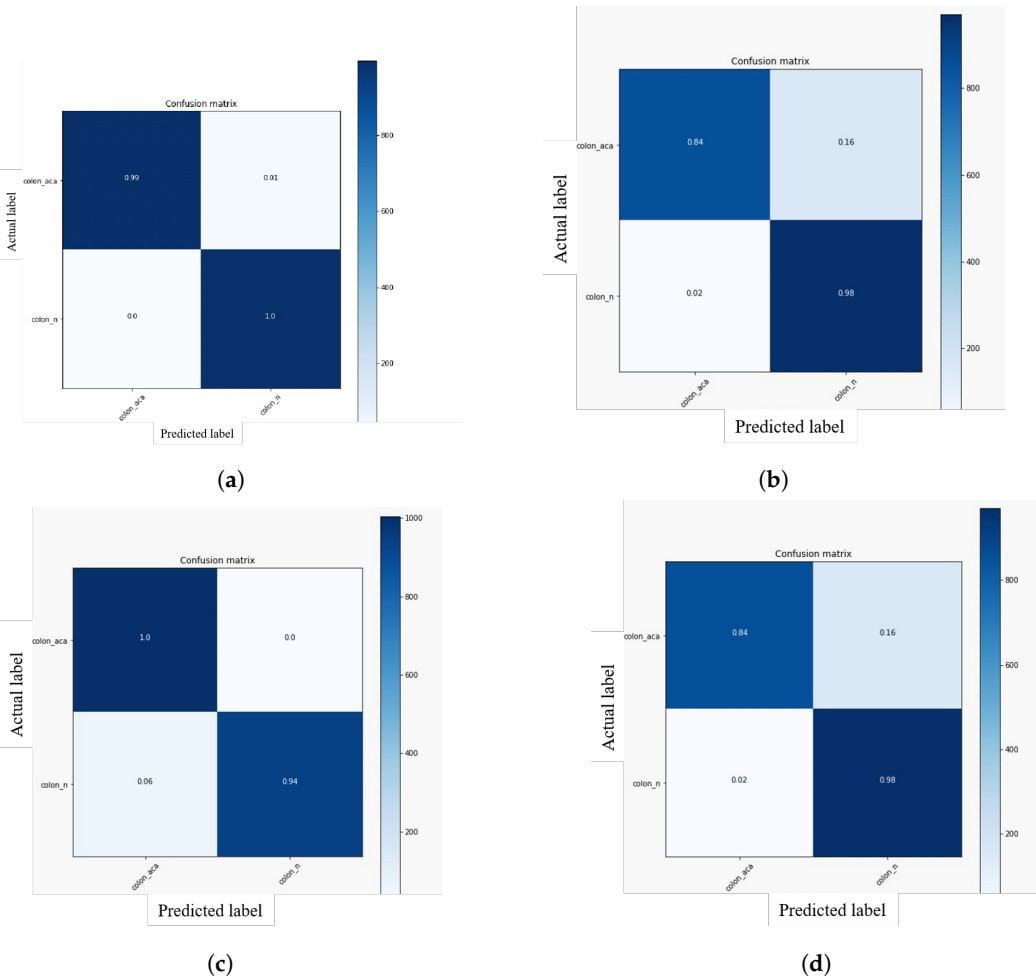

**Figure 5.** Comparison between the confusion matrix of the selected model (**a**) and other models (**b–d**).

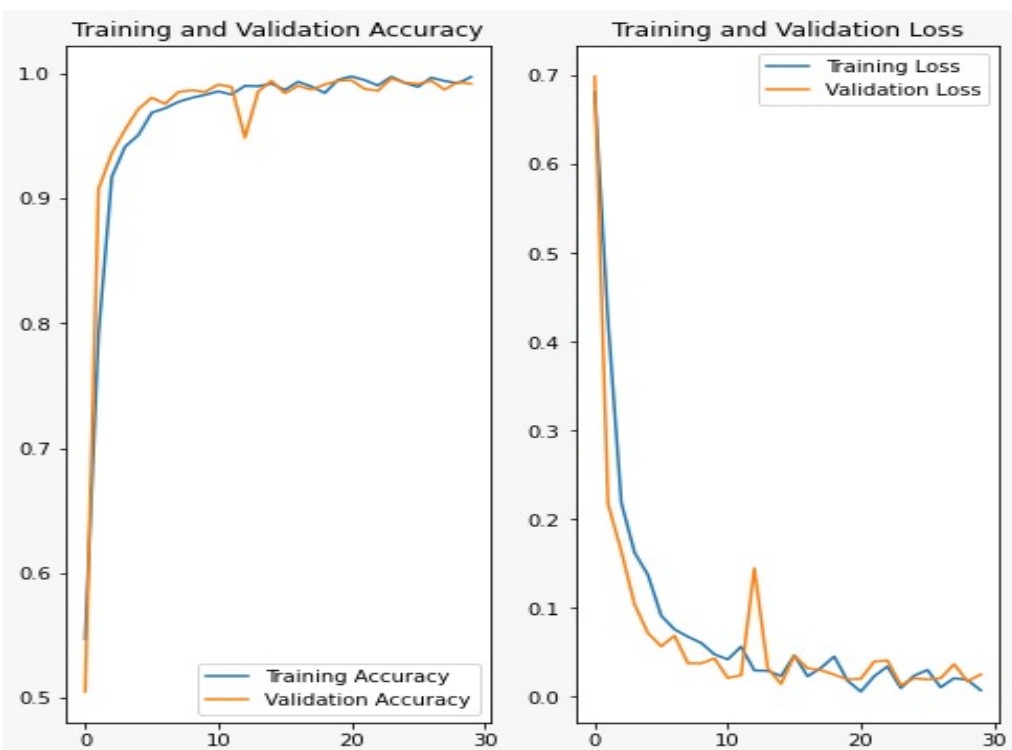

**Figure 6.** Accuracy and loss curves for training data and validation data during 30 epochs.

**Table 1.** Comparison between the seven models (the best model with **bold**).

| Model | # of Layer | Dropout Rate | Total Parameters | Accuracy |
|---|---|---|---|---|
| Model 1 | 14 | 0.5% | 2,031,778 | 56% |
| Model 2 | 16 | 0.5% | 2,097,826 | 49.50% |
| Model 3 | 10 | 0.5% | 7,954,210 | 95.15% |
| Model 4 | 12 | 0.8% | 2,031,778 | 50.50% |
| Model 5 | 10 | 0.8% | 4,063,138 | 98.53% |
| Model 6 | 14 | 0.8% | 7,954,210 | 97.65% |
| **Model 7** | **12** | **0.5%** | **40,631,38** | **99.5%** |

**Table 2.** The performance of the proposed model.

| Accuracy | Precision | Recall | F1-Score | SPE | MCC |
|---|---|---|---|---|---|
| 99.50% | 99% | 100% | 99.49% | 99% | 99% |

Figures 7–10 show the outputs from the considered convolutional layers and max pooling layers corresponding to a sample from the used dataset. We can observe that the deeper the model the more complex the features. As a result, more information can be extracted, which helps the detection process.

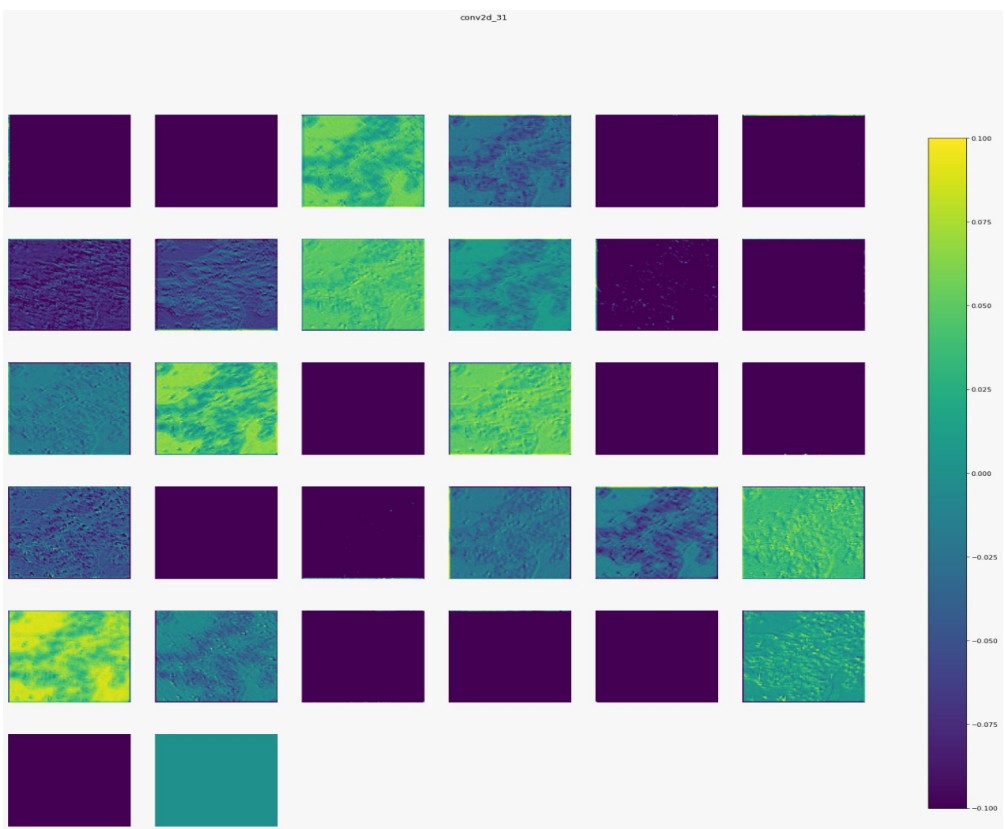

**Figure 7.** The output of the first Convolutional layer.

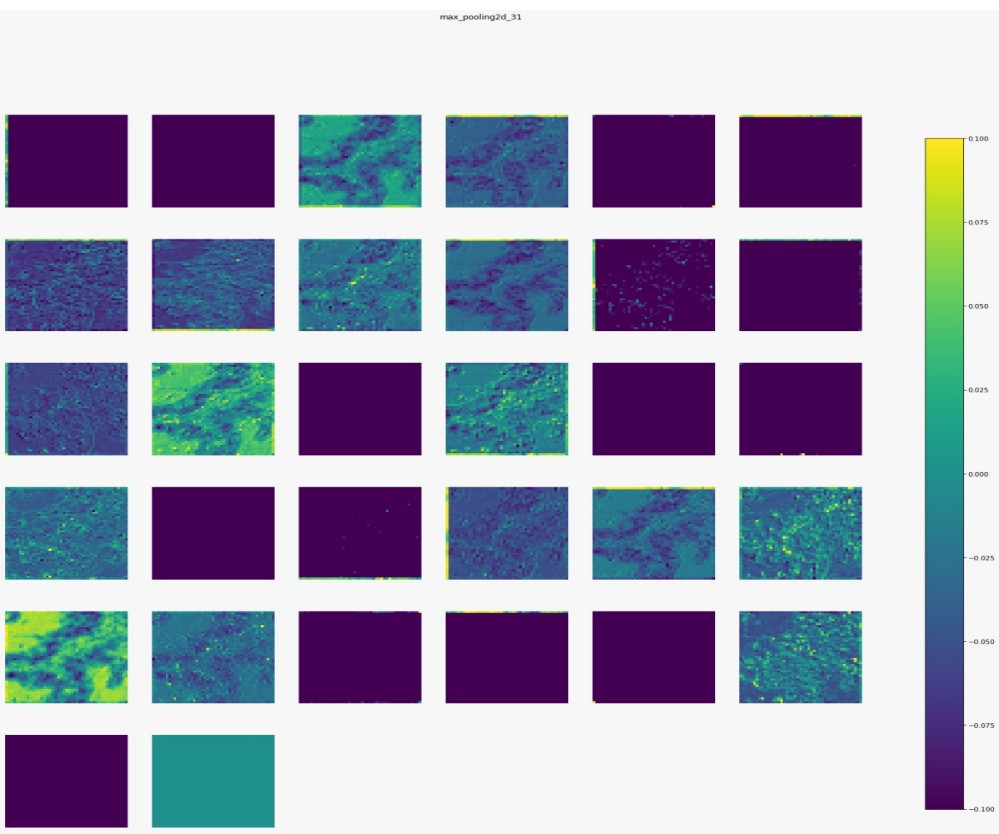

**Figure 8.** The output of the first Max pooling layer.

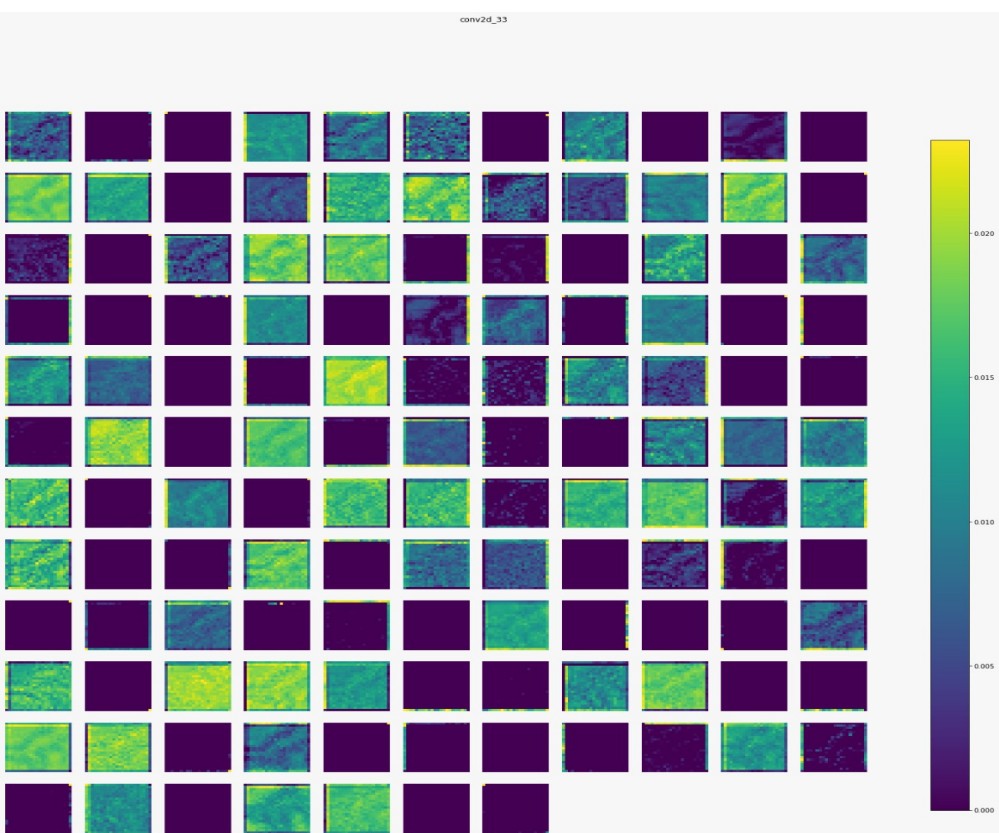

**Figure 9.** The output of the fourth Convolutional layer

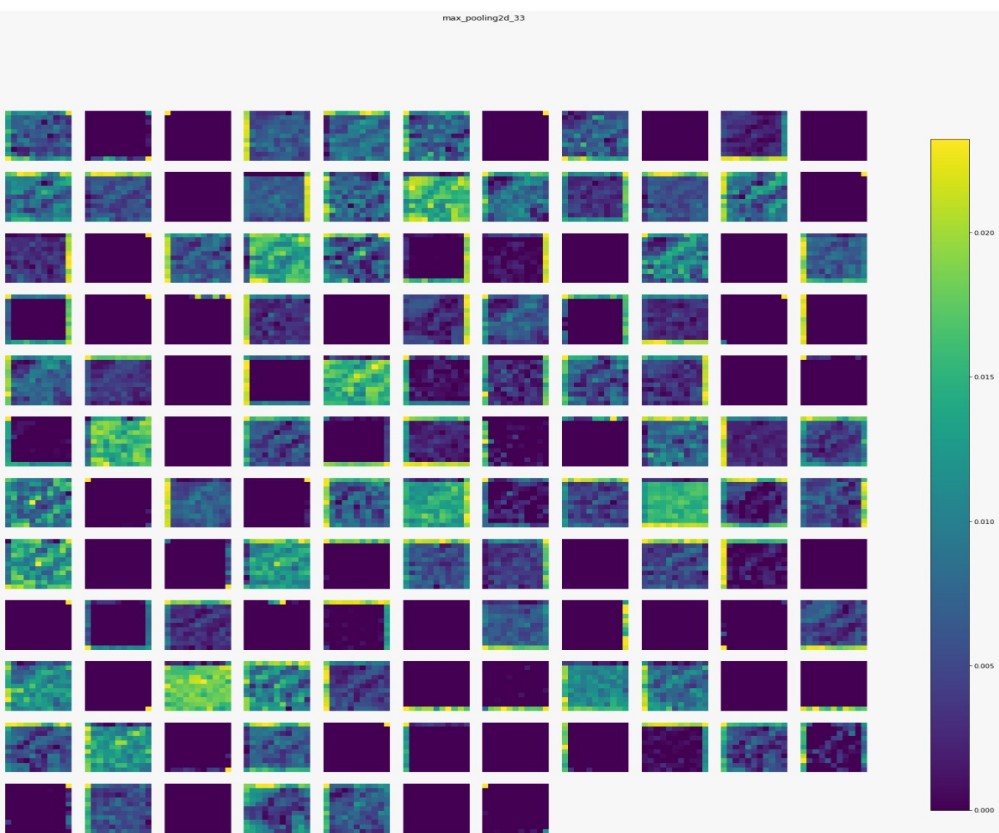

**Figure 10.** The output of the fourth Max pooling layer

Figure 11 shows the distribution of the bias among the 30 epochs. We can observe that the proposed model reaches a zero bias at epoch 19 when the variance equals 0.0599. As a result, our model can fit the training data with low error on testing data, which reduces the overfitting and overcomes the underfitting problems.

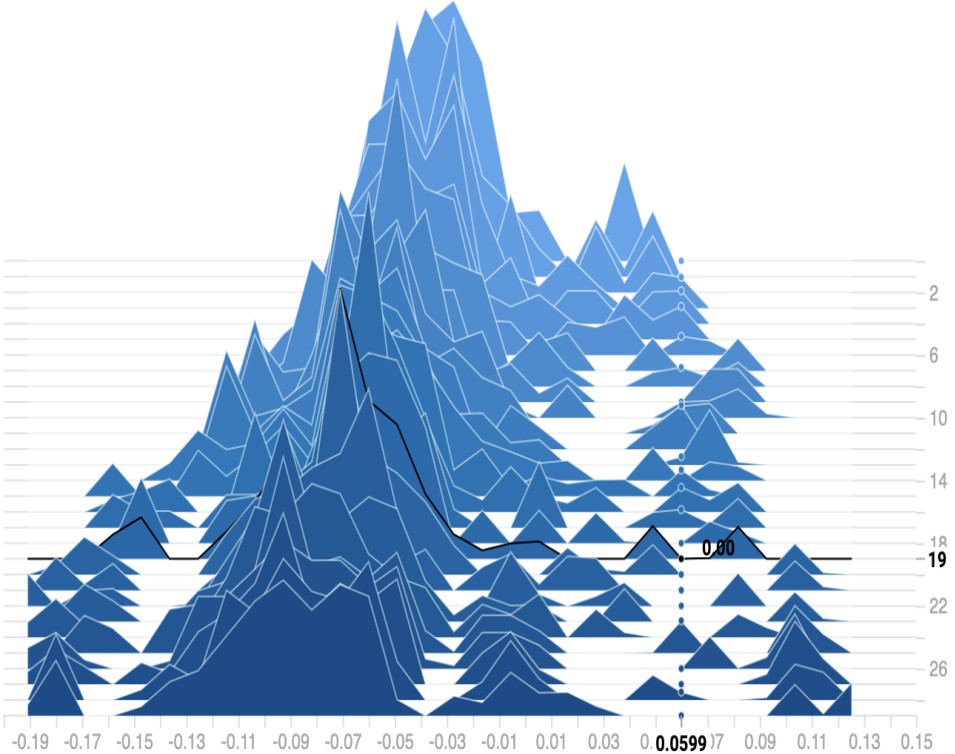

**Figure 11.** Distribution of the bias along the 30 epochs.

Colon cancer detection at the early stages of disease appearance using easy and cheap methods is a key requirement for the prudent application of control measures for this cancer. Therefore, the main goal of this study is to propose a new efficient method for the detection of colon cancer using an easy and cheap technique. The most common approaches for colon cancer detection are certain screening tests such as Sigmoidoscopy, and Colonoscopy. However, these methods need laboratory analysis and technical knowledge, which is expensive and always unavailable. In addition to these conventional techniques, AI has been used to detect colon cancer recently. However, these methods are complex and still suffer from several drawbacks such as time and cost complexity. Therefore, researchers developed methods based on deep learning for colon cancer detection (discussed in Section 2). The comparison of our deep model with these previous models using histopathological images is shown in Table 3.

**Table 3.** Comparison between our deep model and other previous methods on histopathological images.

| Author/Year | Approaches | Accuracy |
|---|---|---|
| Hamida et al., 2021 [9] | Data augmentation and transfer learning | 99.12% |
| Tongacar et al., 2021 [5] | DarkNet-19 model and SVM | 99.69% |
| Yildirim and Cinar et al., 2021 [7] | CNN-based, MA_ColonNET | 99.75% |
| Ohata et al., 2021 [11] | DenseNet169 and SVM | 92.08% |
| Our Method 2022 | Lightweight CNN | 99.50% |

In Table 3, we can observe that our model is more robust and efficient than most previous models. Our model is lightweight compared to other methods as our model consists only of four convolution layers and two dense layers, which are the smallest layer

numbers compared with other models. As a result, reduce the time and cost complexity compared with previous models. Tongaçar [5] obtained high accuracy compared with our model. However this method used a separate classifier (SVM) which increases the complexity and the process time of the method. In addition, the increase in the accuracy was only 0.19% which makes the result of our method is acceptable. Moreover, the method by Yildirim and Cinar [7] is more complex than our model as this method is consisting of 45 layers. Finally, we confirm that a new efficient deep CNN method based on analysis of histopathological images has been confirmed for colon cancer detection.

## 5. Conclusions

The main contribution of this paper is to propose a new lightweight deep learning approach based on CNN for efficient colon cancer detection. The efficiency of the proposed system is analyzed with histopathological images database and is compared with the existing approaches in this field. Results show that our method outperformed most of the previous deep learning approaches for colon cancer detection. The best accuracy, precision, recall, and F1-score of our model are 99.50%, 99%, 100%, and 99.49%, respectively. The proposed method is more robust and more efficient than other previous deep models for the detection of colon cancer. Our application can be useful in some cases where the pathologist who checked the colon images needs to be insured, so our system will help them to reach the right diagnosis. In the future, we will test our deep model on different datasets to see how it performs. In addition, we can employ and combine one of the optimization techniques such asa genetic algorithm with our deep model to select the best features from the extracted deep features.

**Author Contributions:** Conceptualization, A.S.S. and M.H.; methodology, A.S.S. software, A.A.A. and M.S.A.-G.; validation, P.P., and N.F.S. formal analysis, A.S.S.; investigation, A.A.A., M.S.A.-G.; resources, N.F.S.; data curation, P.P.; writing—original draft preparation, M.H.; writing—review and editing, A.S.S.; visualization, A.S.S.; supervision, P.P.; project administration, A.A.A.; funding acquisition, N.F.S. All authors have read and agreed to the published version of the manuscript.

**Funding:** Princess Nourah bint Abdulrahman University Researchers Supporting Project number (PNURSP2022R66), Princess Nourah bint Abdulrahman University, Riyadh, Saudi Arabia.

**Institutional Review Board Statement:** Not applicable.

**Informed Consent Statement:** Not applicable.

**Data Availability Statement:** Not applicable.

**Acknowledgments:** Princess Nourah bint Abdulrahman University Researchers Supporting Project number (PNURSP2022R66), Princess Nourah bint Abdulrahman University, Riyadh, Saudi Arabia.

**Conflicts of Interest:** The authors declare no conflict of interest.

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
