# Peer review of "An Efficient Deep Learning Approach for Colon Cancer Detection"

_applsci, doi:10.3390/app12178450_

Round 1
Reviewer 1 Report
See the attachment.

Author Response
The answers to the comments are attached in the following pdf file.

Reviewer 2 Report
This article “An Efficient Deep Learning Approach for Colon Cancer Detection” by Sakr etc. reports a deep learning approach to analyze input histopathological images for colon cancer detection. There are some questions that need to be addressed before this article can be considered for publication.
1. In Figure 1, it’s better to number the normal images and images with colon cancer, e.g., colon_n_1, colon_n_2, …, colon_aca_1, colon_aca_2, …, so that these images can be clearly referred.
2. The display of Figure 4 is terrible. The texts are too small. It is hard to for the readers to get the key information.
3. In Figure 5, the texts are small and unclear. The label of the Y axis is very blurry.
4. In Table 3, it seems that Tongacar et al. has achieved an accuracy of 99.69% and Yildirim and Cinar et al. has reached 99.75%. In comparison, this paper reports an accuracy of 99.50%. But in the abstract, the authors stated that “The result analysis demonstrates that the proposed deep model for colon cancer detection provides the result with higher accuracy of 99.50% which is considered the best accuracy compared with other deep learning approaches”. It seems that 99.50% cannot be considered to be the best.
5. The grammar of the whole manuscript needs to be checked before publication.
Author Response

(The authors gave the same response as above.)

Round 2
Reviewer 1 Report
No questions.